# Visual orientation discrimination skills are tightly linked with specific aspects of human intelligence

Kyriaki Mikellidou[1,2,3]*, Nefeli Lambrou[1], Ellada Georgiou[1], Marios Avraamides[1]

**1** Department of Psychology, University of Cyprus, Nicosia, Cyprus, **2** Department of Management, University of Limassol, Limassol, Cyprus, **3** Department of Neuroscience, Psychology, Pharmacology and Child Health, University of Florence, Florence, Italy

* kmikellidou@gmail.com

## Abstract

We investigate the notion that basic visual information is acting as a building block for more complex cognitive processes in humans. Specifically, we measured individual visual orientation discrimination thresholds to report significant correlations against the total standardised intelligence quotient (IQ), verbal-IQ and non-verbal IQ scores evaluated using the Wechsler Abbreviated Scale of Intelligence Second Edition (WASI-II) test battery comprising Verbal Reasoning, Block Design, Similarities and Matrix Reasoning subtests (N = 92). A multiple linear regression analysis showed that participants' performance in our visual discrimination task, could be explained only by individual scores in Verbal Reasoning which quantifies the ability to comprehend and describe words and Matrix Reasoning, which evaluates general visual processing skills including abstract and spatial perception. Our results demonstrate that low-level visual abilities and high-level cognitive processes are more tightly interwoven together than previously thought and this result could pave the way for further research on how cognition can be defined by basic sensory processes.

## Introduction

Past research has documented a link between intelligence and low-level visual perception, with higher Intelligence Quotient (IQ) associated with faster and more accurate perceptual judgements. This has been demonstrated with a variety of sensory discrimination tasks, including pitch and colour discrimination [1], motion discrimination [2, 3] and contrast discrimination [4]. An observed correlation between nerve conduction velocity in the visual pathway running from the retina to the primary visual cortex via the lateral geniculate nucleus is correlated with intelligence is also in support of this notion [5].

Perceptual tasks typically correlate well with IQ, usually between r = 0.2 and r = 0.4 [6], but spatial suppression, in particular, appears to have a stronger association with IQ scores. Specifically, Melnick et al. [3], reported strong correlations (r = 0.64; p = 0.02 and r = 0.71; p = $10^{-9}$ respectively, 95% CI) between IQ and the Suppression index (SI), which was used to index the inhibitory mechanisms that make motion-selective neurons less responsive to large,

**Data Availability Statement:** Our full data set is available on OSF: https://osf.io/gxjtq/files/ osfstorage/64c234e1684bcc00562042e5.

**Funding:** This project has received funding from and the Marie Skłodowska-Curie programme Grant agreement No 797603 (Pheripheality; K.M., N.L., M.A). The funders had no role in study design, data collection and analysis, decision to publish, or preparation of the manuscript.

**Competing interests:** The authors have declared that no competing interests exist.

background-like motion patterns [7]. SI is thought to carry information about our ability to suppress irrelevant information and the speed with which fast relevant information can be processed. This finding, documenting that the accuracy and speed processing of low-level visual properties predicts intelligence, is in line with Galton's conjecture that intelligence and simple sensory discriminations operate via common neural processes [8]. In further support of this idea, Rademaker et al. [9] showed that population-level response patterns in the early visual cortex represent the contents of working memory alongside new sensory inputs. Indeed, when participants were distracted in the study of Rademaker et al. [9], disruptions of mnemonic information in early visual cortex and decrements in behavioural recall were observed.

Additional factors, such as participants' motivation and their state of alertness can also exert an influence on psychophysical task performance. For this reason, Troche et al. [10] argued that observed correlations between psychophysical task performance and a related construct such as intelligence could be driven by an irrelevant source of variance, an issue sometimes referred to as the impurity problem [11]. To tackle this issue, Troche et al. [10] combined traditional structural equation modeling and the fixed-links modelling approach to try and reveal the unique effects of mental speed and spatial suppression on psychometric intelligence. In contrast to Melnick et al., [3], they failed to find a link between spatial suppression and mental abilities, although stimuli and apparatus differences could be the underlying cause of this discrepancy.

As Tadin et al. [12] argued, rapid processing of relevant information and suppression of redundant and less informative signals are crucial properties of any system operating on information that exceeds its processing capacity. This ties into a broader scheme whereby information suppression in any brain system that operates on exceeding capacity, such as working memory, is crucial for its efficiency and accuracy in both sensory discrimination and higher-order reasoning. A question that arises from this past research is whether the tight coupling observed between IQ and visual motion suppression mechanisms, which are thought to reflect the receptive field properties of centre-surround neurons in the motion-sensitive brain area MT, can be also observed with an orientation discrimination task.

Considering that orientation discrimination relies heavily on specialised orientation-tuned neurons in the primary visual cortex [13], such a task would allow us to evaluate one of the most common visual mechanisms that could be driving the visual skills that are commonly tested in intelligence tests. Previous research examining orientation sensitivity using line stimuli in typically developing young boys (7–15 years), showed a negative correlation, which marginally failed to reach significance [$R(34) = -0.31$, $p = 0.06$, $N = 38$] [14]. Although these results could point towards a possible relationship between IQ and orientation discrimination thresholds, the young age of the participants, the gender and the relatively small sample size do not allow us to draw firm conclusions from these findings. In addition, there is no information regarding the relationship of orientation sensitivity with verbal subtests (i.e. Verbal Reasoning, Similarities) and non-verbal subtests of the intelligence quotient (i.e. Block Design and Matrix Reasoning).

Thus, the aim of the present study is to determine whether performance on a visual orientation discrimination task can be linked with overall intelligence quotients and specific subtests. In the study, orientation discrimination thresholds were measured as a function of the Just Noticeable Difference in degrees, i.e., how much difference in degrees of orientation is needed for a participant to realise that two stimuli have a different orientation. Intelligence was measured using the Wechsler Abbreviated Scale of Intelligence Second Edition (WASI-II) test battery comprising Verbal Reasoning, Block Design, Similarities and Matrix Reasoning subtests. To control for the effects of age and education in IQ scores, our sample was limited to young adults, all with university-level education.

## Methods

### Participants

Experimental procedures were approved by the Cyprus National Bioethics Committee (Β Π 2018.01.183) and are in line with the Declaration of Helsinki abiding to ethical standards that promote and ensure respect for all human participants, while protecting their health and rights. Research practices conformed to generally accepted scientific principles and were all based on a thorough knowledge of relevant scientific literature. All participants provided written informed consent prior to participating in the experiment.

We tested a total of 106 participants (62 female; age range 18–36; mean age = 22.5). Inclusion criteria involved normal or corrected to normal vision, no neurological disorders, no learning disabilities and currently in or already possessing university-level education. They were recruited from a participant pool with students from the University of Cyprus, via the internet, and by word of mouth. Participants received either a monetary reimbursement of €10/hour or course credit in exchange for their participation. In total, 14 participants were excluded from the analyses; two had vision that was not normal (S1 wore permanent contact lenses, S25-anisometropia), one did not bring their prescription lenses with them (S46) and failed to inform the experimenter prior to the beginning of the experiment. Six participants were excluded because no psychometric function could be fitted to their data (S2, S3, S8, S69, S70, S98), one participant was excluded due to photosensitivity, with their eyes becoming watery during the psychophysics session and being unable to perceive the stimuli normally (S42), two participants were considered outliers i.e. more than three standard deviations away from the mean (S38: JND = 25.9, S77: JND = 24.1), and two participants were excluded because they did not complete all parts of the experiment (S52, S86). Therefore, the final analysis included data from 92 participants (52 female; mean IQ = 98, s = 10.9).

### Materials

**Orientation discrimination task.**   We used a typical orientation discrimination test, used widely by visual perception researchers. Presentation time was 500ms for each grating because our participants were inexperienced with psychophysical tests and found it challenging to carry out the task with shorter presentation times. To evaluate orientation discrimination abilities, participants were asked to fixate their gaze on a small central fixation square for the whole duration of the experiment and compare the orientation of two sequentially presented Gabor patches (diameter = 5 degrees) at an eccentricity of 7.5 degrees (Fig 1). Stimuli were placed at this eccentricity to avoid microsaccadic eye movements which could potentially hinder visual perception. The spatial frequency of the Gabor patches was 1.5 cycles/degree, presented at 100% contrast. The method of constant stimuli was used with the standard stimulus oriented at 45° compared with the variable stimulus ranging between 30° and 60° in steps 3°. The order of presentation of the two gratings (standard and variable) was random in each trial. After the presentation of the stimuli, participants indicated which of two sequentially presented Gabor patches was oriented further clockwise. We used an ASUS Sonic Master laptop with Intel® HD Graphics 4000 card, Intel Core i5, 4GB memory. Stimuli were presented on a calibrated 23-inch Full HD LCD monitor subtending 26° (horizontal) by 14.5° (resolution: 1920 x 1080 pixels, frame rate = 60Hz). Stimuli were generated with MATLAB (the MathWorks, Natick, MA) using routines from the Psychtoolbox [15–17]. Responses were collected via a standard keyboard connected via USB to a PC yielding a temporal resolution of 4 milliseconds.

**Standardized Wechsler Abbreviated Scale of Intelligence (WASI-II) battery of tests.**
WASI-II measures the intellectual ability of people aged between 6–89 years old. The

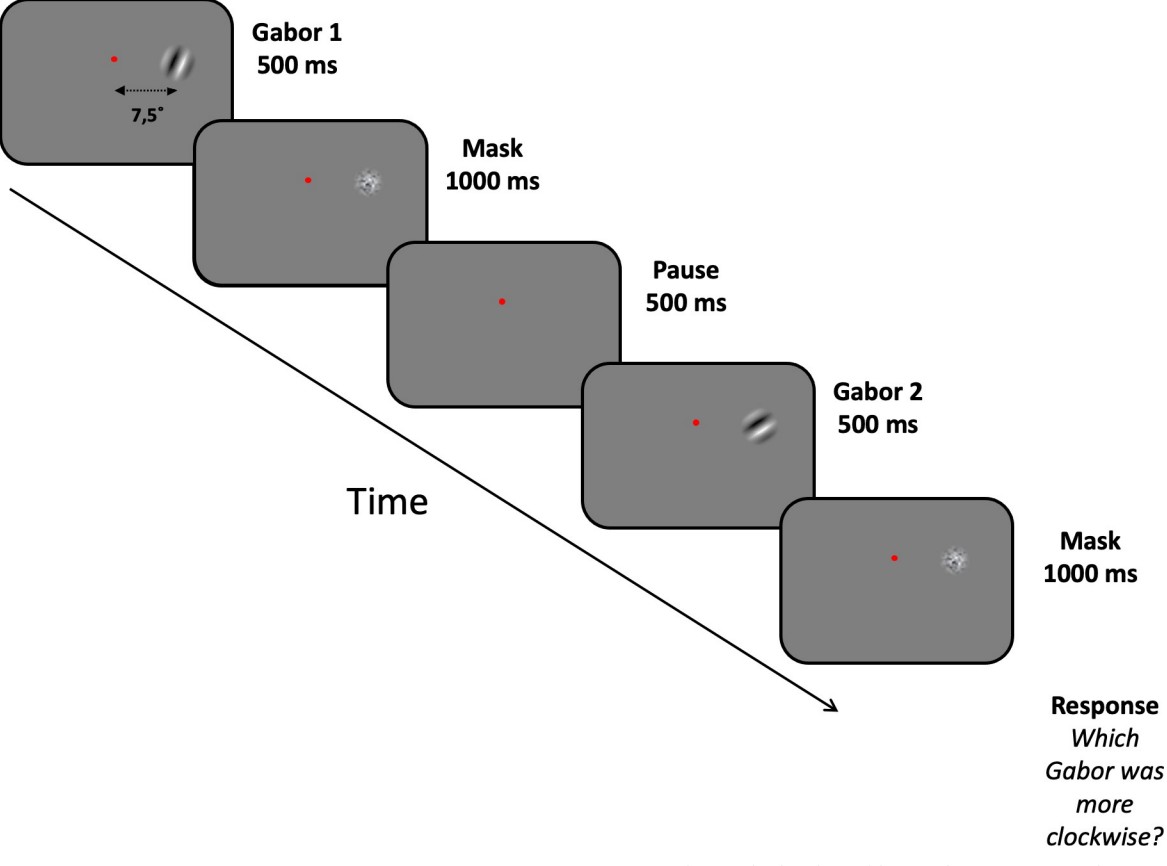

**Fig 1. Timeline and stimuli of the two-AFC orientation discrimination task.** The standard and variable stimulus were presented sequentially for 500ms each, 7.5 degrees to the right of fixation which had to be maintained throughout the course of the experimental session. Following the presentation of each one, a mask appeared to eliminate any aftereffects which could affect orientation discrimination. At the end of each trial, participants were asked to indicate which grating (the first or second) was more clockwise.

standardized scaled scores of WASI-II range between 50–160. We used a standardised WASI-II for the Greek-Cypriot population in Cyprus in Greek [18], the native language our participants. To administer the WASI-II we used an *online answer sheet*, *the book of stimuli*, *a set of 9 cubes*, and a *stopwatch* as required. The four subscales were administered in a random sequence, alternating between verbal and non-verbal subscales: *(A) Verbal reasoning*–evaluates the ability to describe verbally a specific word *(B) Block Design*–requires recreating a pattern using bi-coloured cubes within a time limit *(C) Similarities*–measures the ability to describe verbally how two words are related (e.g., "how are strawberries and pears similar?"), *(D) Matrix Reasoning*–examines how well one can indicate the missing piece of a pattern amongst five options. The average time needed for the WASI-II to be completed is 45 minutes.

**Procedure.** Participants were asked to come to the lab well-rested. Upon arriving, they were asked to read an information sheet about the experiment and then to sign an informed consent form. Participants were seated behind a desk and completed the four subtests of the Standardized WASI-II IQ test in Greek that entailed describing words, constructing patterns using wooden cubes, finding similarities between two words, and indicating which picture matched a given pattern.

Either before or after the completion of the WASI-II IQ test, participants performed a two-alternative forced choice (2-AFC) orientation judgement task aimed at measuring individual sensitivity in an orientation discrimination task. Participants performed three runs (55 trials

each run) of the task. The first run was considered practice and was not used in subsequent data analysis.

**Data analysis.** We used MATLAB (R2019b) and JASP 0.16 for data analysis. We calculated correlations between different variables using Pearson's r and carried out multiple linear regression; neither of these require the assumption of normality. To carry out a comparison between students of more theoretical and practical degrees we use the non-parametric Mann-Whitney U. To evaluate normality, we used the Shapiro-Wilk normality test.

## Results

Our final sample consisted of 92 participants. The average IQ was 98 (SD = 11), while the average normalised score for Verbal Reasoning was 44.7 (SD = 7.7, Range = 28–60), for Block Design 53.1 (SD = 6.7, Range 38–66), for Similarities 46.6 (SD = 7.1, Range = 31–61) and for Matrix Reasoning 52.2 (SD = 7.7, Range = 34–68). Psychometric functions were computed using 110 trials of the orientation discrimination task for each participant which were fitted with cumulative Gaussians using maximum-likelihood estimation (100 bootstrap repetitions; see Appendix I in S1 Appendix for examples). From this, the JND was computed in degrees of orientation and was plotted as a function of the WASI-IQ scores. The average JND was 6.77 (SD = 3.00) degrees of visual angle. The Shapiro-Wilk was used to check the assumption of normality and we found that this was violated for JND (p<0.001), Similarities (p = 0.008) and Matrix Reasoning (p = 0.033), so we proceeded with statistical tests that are independent of this assumption.

Fig 2A shows a significant negative correlation between IQ scores and JND (Pearson's r = -0.39, $R^2$ = 0.16, p<0.001), indicating that participants with a higher intelligence quotient have also more precise orientation discrimination skills (i.e. smaller JNDs). As shown in Fig 2B and 2C, we also found negative correlations between JND and standardized Verbal IQ scores computed using the Verbal Reasoning and Similarities scores (Pearson's r = -0.33, $R^2$ = 0.11, p = 0.001) and standardized Non-Verbal IQ scores computed using Block Design and Matrix Reasoning scores (Pearson's r = -0.36, $R^2$ = 0.13, p<0.0021).

We used a multiple regression analysis with a stepwise method to test if any of the four factors can predict participants' performance in our visual discrimination task, as defined by the JND. The results of the regression analysis using normalised scores indicated that Verbal Reasoning and Matrix Reasoning explained 15% of the variance of visual orientation discrimination skills ($R^2$ = .15, r = 0.387, F(2,89) = 7.84, p < .001). The other two subtests (Block Design, Similarities) were considered, but not included in the final model. Table 1 shows that performance Matrix Reasoning significantly predicted JND (unstandardised β = -0.092, standardised β = -0.240, p = .023), as did performance in Verbal Reasoning (unstandardised β = -0.090, standardised β = -0.234, p = .026). Although the normalised scores from all four subtests were found to be correlated between them and with JND (see Appendix II & III in S1 Appendix), only two of them contribute significantly to explaining visual discrimination abilities. We report that Cook's distance calculations revealed no highly influential participants affecting the regression model (i.e., all values <0.2, with the threshold usually set at <1). Taken together, our results show that Matrix Reasoning which measures visual processing and abstract, spatial perception and Verbal Reasoning which measures verbal language comprehension, reasoning and logic through the understanding of language can reliably explain visual discrimination abilities.

On a side note, we also assessed whether scaled scores on each subtest varied in accordance with the type of degree each participant studied towards or had already completed. We compared performance across the four subtests for participants in two groups: those with degrees

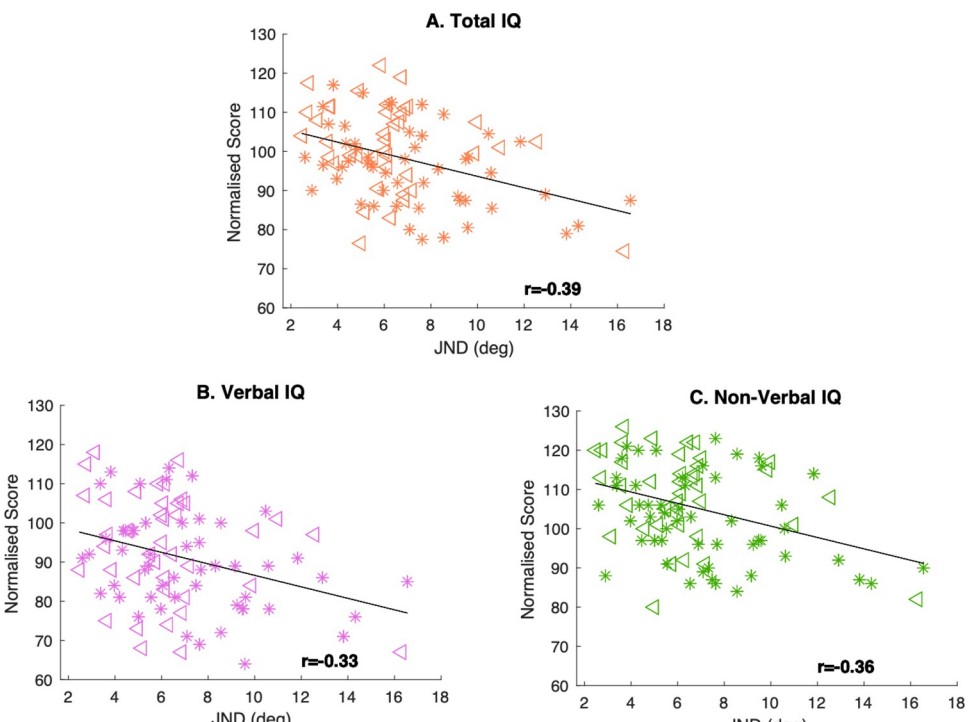

**Fig 2. Individual normalised scores for total, Verbal and Non-verbal IQ correlated against JND in degrees of orientation (N = 92).** Triangles indicate males, stars indicate females. A. Total Intelligence Quotient (Total IQ) as measured by WASI-II plotted against Just Noticeable Difference (JND) in degrees of orientation. The linear regression model shows a negative correlation between the two variables ($R^2$ = 0.16, p<0.001). B. Scaled Verbal IQ is independently and negatively correlated with JND ($R^2$ = 0.11, p = 0.001). C. Scaled non-verbal IQ has an independent and even stronger negative correlation with JND ($R^2$ = 0.13, p<0.001).

of a theoretical nature (*N = 58*, i.e., Psychology, Education Studies, Philosophy, Literature, Languages, History, Archaeology) and those with degrees of a more practical nature (*N = 33*, i.e., Engineering, Computer Science, Economics, Medicine, Architecture, Biotechnology). We observed significantly higher scores in the two non-verbal tests for the practical group compared to the theoretical group. Specifically, in Block Design participants with degrees of a more practical nature ($M_P$ = 55.7, $SD_P$ = 6.0) had significantly higher normalised scores compared to those with degrees of a theoretical nature ($M_T$ = 51.8, $SD_T$ = 6.7; Mann-Whitney U = 1264, p = .011, equal variances assumed using Levene's test for homogeneity of variance, F (1) = 1.429, p = 0.235). The same pattern of results was observed in the Matrix Reasoning subtest as participants with degrees of a more practical nature ($M_P$ = 54.5, $SD_P$ = 8.2) had significantly higher normalised scores compared to those with a theoretical background ($M_T$ = 50.9, $SD_T$ = 7.2; Mann-Whitney U = 1242, p = .018, equal variances assumed using Levene's test for homogeneity of variance F(1) = 0.154, p = 0.695). No significant differences were found for the

**Table 1.**

|  | Unstandardised | St. Error | Standardised | t | p |
|---|---|---|---|---|---|
| **Intercept** | 15.612 | 2.262 |  | 6.903 | < .001 |
| **Matrix Reasoning** | -0.092 | 0.040 | -0.240 | -2.317 | 0.023 |
| **Verbal Reasoning** | -0.090 | 0.040 | -0.234 | -2.263 | 0.026 |

two verbal subtests (Verbal Reasoning: $M_P$ = 44.8, $SD_P$ = 8.8; $M_T$ = 44.8, $SD_T$ = 7.1, Mann-Whitney U = 1002, p = .713; Similarities: $M_P$ = 47.5, $SD_P$ = 7.1; $M_T$ = 46.0, $SD_T$ = 7.1, Mann-Whitney U = 1069.5, p = .354). We observed no significant difference in orientation discrimination performance (measured using the JND) between the two groups (Practical: $M_P$ = 6.2 degrees of orientation, $SD_P$ = 2.7 degrees of orientation vs. Theoretical degrees: $M_T$ 7.1 degrees of orientation, $SD_T$ = 3.1 degrees of orientation, Mann-Whitney U = 795.5, p = .184)

## Discussion

The aim of this study was to investigate whether basic sensory information acts as a building block for much more complex cognitive processes as evaluated by intelligence testing. The novelty lies in the visual task we have used which evaluates one of the most common abilities of any visual system: orientation discrimination (for a review see [19]). The significant correlations we found between overall intelligence scores and orientation discrimination abilities, show that participants with higher IQ scores have reduced JNDs (i.e., more accurate orientation discrimination skills). This result is in line with previous work reporting a negative—albeit marginally non-significant—correlation between IQ and discrimination thresholds in young boys [14]. Non-verbal skills, evaluated by combining scores from the Matrix Reasoning and Block Design subtests and verbal skills evaluated by combining scores from the Verbal Reasoning and Similarities subtests, showed significant negative correlations with JND as well. Although we do understand that correlation does necessarily mean causation, we take these correlations as evidence of the fundamental role that vision plays in high-level human functions. Finally, linear multiple regression model showed that the Verbal Reasoning and the Matrix Reasoning subtests can independently explain part of the variance we observed in individual orientation discrimination skills.

We also report that individuals with a more practical educational background exhibit a better performance in the two non-verbal subtests of WASI-II. The educational background was defined as "practical" if during the studies individuals spend a significant amount time dealing with mathematical calculations and/or manual work with or without tools. It is important to note that although our study does not directly evaluate maths performance, this result is in line with previous research showing a link between non-verbal fluid intelligence and general maths performance [20]. We are not able to tell whether individuals develop their non-verbal skills during the course of their studies or whether they follow such a more "practical" path because they have superior non-verbal skills to start with.

### Orientation discrimination in the brain

Orientation discrimination abilities are thought to be defined by early visual brain areas and mainly the primary visual cortex (V1) which is traditionally believed to operate in a purely sensory manner by receiving direct retinal input from the eyes and constructing an accurate retinotopic map of the visual world. Rademaker et al. [9] suggested that, in reality, V1 may potentially possess co-existing representations of both sensory and mnemonic information, running via a 'local comparison circuit'. Separable bottom-up and top-down inputs could theoretically support the coexistence of multiple simultaneous representations. In one experiment, reported by Rademaker et al., population-level response patterns in early visual cortex were found to represent the contents of working memory alongside new sensory inputs. In a second experiment, Rademaker et al. [9] showed that when participants got distracted, both disruptions of mnemonic information in early visual cortex and decrements in behavioural recall were observed. Therefore, it can be inferred that information already present in working memory (from early retinotopic cortex) is maintained even after new sensory inputs are processed into visual working memory. However, salient and distracting information can

negatively impact recall, suggesting that early visual areas actively participate in both sensory and mnemonic processing. In support of this notion, Vredeveldt, Hitch, and Baddeley [21] showed that closing one's eyes reduces cognitive load and aids memory recall. In their study, they exposed participants to a violent video clip and later to different types of distraction during a witness interview (blank screen—control, eyes closed, visual distraction, auditory distraction). Recall was significantly better when distraction was minimal, providing evidence that closing the eyes reduces input to V1, and as a result cognitive load.

## Inhibitory processes affecting higher level functions

Our results appear to agree with the broader literature regarding sensory discrimination and intelligence. The significant role of Matrix Reasoning as a factor explaining JND in our multiple linear regression model comes as no surprise as the subtest relies on solid visual skills including classification, spatial abilities, knowledge of part–whole relationships and perceptual organization. These results agree with Melnick et al.'s [3] conclusions discussed in the introduction, that accuracy and speed of processing of low-level visual properties may predict higher intellect in individuals. Their results suggest that visual suppression mechanisms play a crucial role in perception by allowing perceptual systems to efficiently process vast amounts of sensory input [22], in addition to an analogous role in intelligent cognition by contributing to improved neural efficiency [23]. Our results go a step further to show that orientation discrimination, a visual skill all animals possess, can be indicative of overall competence skills in humans, evaluated using the WASI-II battery of tests.

Neural suppression efficiency could provide a mechanistic explanation for the SI–IQ link, through multiple inhibitory processes which have been found to strongly predict IQ scores [24], such as attentional and working-memory control [25]. Cook et al. [4] reported that higher levels of cortical gamma-aminobutyric acid (GABA) in the human primary visual cortex were associated with better performance in the Matrix Reasoning IQ subtest ($r = 0.83$, $p = 0.0054$). Further, Edden and colleagues [26] added that neuronal inhibition is mediated by GABAergic interneurons, possibly influencing performance in orientation judgement tasks. Thus, GABA constitutes a plausible candidate neural substrate for Melnick et al.'s proposal about the link between SI and IQ in the suppression of irrelevant information.

The significant role of Verbal Reasoning as a factor explaining JND in our multiple linear regression model could therefore be attributed to information suppression abilities: participants who required fewer degrees of orientation difference to realise that two successive Gabor patches were different, can probably 'shed' unnecessary information more effectively in general. The importance of vision in language acquisition through reading and writing in neurotypical and sighted humans could also explain this relationship. Bedny et al., [27] showed that cortical regions in the occipital lobe of blind human adults that are typically specialised in visual processing in sighted adults, appear to be taking on language processing as a result of early experience. Such cortical reorganisation has been widely reported in the literature and it is believed to facilitate the efficient processing of sensory input by available and capable brain areas [28], in this case the occipital cortex of blind individuals. This would suggest that the occipital cortex has the capacity to process language at a supra-modal level. However, the explanation could be much simpler: participants who were able to comprehend task instructions more efficiently because of superior verbal reasoning skills, could have an advantage solely because of this.

## Working memory and the "g" factor

Given that participants were asked to compare two Gabor gratings presented within a two-second window, the task administered to the participants could be considered a short-term

memory task. Thus, one possibility is short-term memory being at the heart of the correlations observed between JND and the four intelligence subtests (see Appendix III in S1 Appendix). Individuals who can retain information in their short-term memory more efficiently may perform better in a variety of tasks. Another possibility is that results could indicate a 'positive manifold', where many kinds of cognitive tests (i.e., arithmetic and vocabulary) are all positively correlated with each other [29]. In fact, by examining various results from general intelligence tests (from simple sensory discrimination to reaction times of highly complex problem solving), Jensen [30] suggested that the degree of correlation between them can be boiled down to physiological factors such as brain size, brain nerve conduction velocity and brain glucose metabolic rate. Indeed, Colom and colleagues [31] found that participants scoring higher in intelligence tests had greater grey matter volume in all brain lobes, suggesting that perhaps "g" relates to distributed networks across the whole brain.

Further into this, Kovacs and Conway [32], introduced the Process Overlap Theory, a unified account about the general factor of intelligence "g", which states that during cognitive tests, executive processes are tapped into in an overlapping manner. According to this, there are distinct within-individual processes tapped by different test items which might belong to different cognitive domains. Such processes could give rise to apparent correlations between intelligence subtests and orientation discrimination abilities, while in fact the true correlation is driven by central executive processes and the ability to allocate cognitive resources to complete a specific task. This idea of a central integrative system of coordinating between processes [33], describes a mechanism through which information could be maintained in a readily accessible state but could also engage in concurrent processing, while still having access to long-term memories. Later, Baddeley [34] developed a more nuanced model of memory and clarified that communication between each subsystem is based on the idea that that such transformations critically rely on the central executive. Interestingly, Tsukahara et al. [35] showed that attention control fully mediates the relationships of working-memory capacity to sensory discrimination, evaluated with two visual tasks (circle and line size discrimination) and two auditory tasks (pitch and loudness discrimination). However, in our study, only Verbal Reasoning and Matrix Reasoning were found to explain the variance observed in orientation discrimination abilities across our sample, suggesting that our results go beyond attentional allocation or working memory capacity.

To reduce this working memory component, future work could use another orientation discrimination task with reduced working memory demands, maybe by using a single stimulus and asking participants to judge whether it appears oriented more leftwards or rightwards. If orientation discrimination abilities evaluated in this way still maintain a significant correlation with IQ, this could reinforce further the notion that low-level skills are truly associated with specific high-level functions.

Overall, our results provide evidence that low-level visual abilities and high-level cognitive processes are more tightly interwoven than previously thought. This is of great importance, as basic vision research is often overlooked by researchers who focus on higher-level and often multisensory human abilities, assuming that it is difficult to associate basic visual abilities with more advanced human skills.

## Supporting information

**S1 Appendix.**
(DOCX)

## Author Contributions

**Conceptualization:** Kyriaki Mikellidou.

**Formal analysis:** Kyriaki Mikellidou.

**Investigation:** Nefeli Lambrou, Ellada Georgiou.

**Methodology:** Kyriaki Mikellidou.

**Project administration:** Kyriaki Mikellidou.

**Resources:** Marios Avraamides.

**Supervision:** Marios Avraamides.

**Writing – original draft:** Kyriaki Mikellidou, Nefeli Lambrou, Ellada Georgiou.

**Writing – review & editing:** Kyriaki Mikellidou, Nefeli Lambrou, Marios Avraamides.

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
