## [Decision Letter · Decision Letter 0]

26 Apr 2023

PONE-D-22-29542Visual orientation discrimination skills are tightly linked with specific aspects of human intelligencePLOS ONE

Dear Dr. Mikellidou,

Thank you for submitting your manuscript to PLOS ONE. After careful consideration, we feel that it has merit but does not fully meet PLOS ONE’s publication criteria as it currently stands. Therefore, we invite you to submit a revised version of the manuscript that addresses the points raised during the review process.

We look forward to receiving your revised manuscript.

Kind regards,

Nicola Megna, M.D.

Academic Editor

PLOS ONE

“This project has received funding from and the Marie Skłodowska-Curie programme Grant agreement No 797603 (Peripheality; K.M., N.L., M.A).”

Please state what role the funders took in the study.  If the funders had no role, please state: "The funders had no role in study design, data collection and analysis, decision to publish, or preparation of the manuscript.

“This project has received funding from and the Marie Skłodowska-Curie programme Grant agreement No 797603 (Peripheality; K.M., N.L., M.A).”

We note that you have provided funding information. However, funding information should not appear in the Acknowledgments section or other areas of your manuscript. We will only publish funding information present in the Funding Statement section of the online submission form.

“This project has received funding from and the Marie Skłodowska-Curie programme Grant agreement No 797603 (Peripheality; K.M., N.L., M.A).”

Reviewers' comments:

Reviewer's Responses to Questions

**Comments to the Author**

1. Is the manuscript technically sound, and do the data support the conclusions?

Reviewer #1: Yes

Reviewer #2: Yes

2. Has the statistical analysis been performed appropriately and rigorously? 

Reviewer #1: Yes

Reviewer #2: Yes

3. Have the authors made all data underlying the findings in their manuscript fully available?

Reviewer #1: Yes

Reviewer #2: Yes

4. Is the manuscript presented in an intelligible fashion and written in standard English?

Reviewer #1: Yes

Reviewer #2: Yes

5. Review Comments to the Author

Reviewer #1: Evaluation

In this paper Mikellidou et al. investigate whether a basic visual function, such as visual orientation discrimination, can be correlated with high-order cognitive abilities measured through the Intelligent Quotient (IQ). The authors found a negative correlation between these measures, specifically lower precision in visual orientation discrimination was associated with lower level in particular subtests of Verbal and Non-Verbal IQ. Finally they conclude that these association are not causative but they demonstrate the relationship between basic sensory processing that could in some way influence the develop of individual cognitive factors .

The study appears to have been well executed and the writing is clear and communicates the main points well. I really appreciated the clarity of the introduction. I think that some points should be addressed or clarified before the study is suitable for publication.

Methods

# Visual Discrimination task : Is the position of the standard stimulus randomized across different trials, in order to avoid sequential order effects?

# why the authors choose to present the stimuli in periphery rather than in fovea? This for my opinion would be a way to avoid eye movements of the subjects.

#Results

# in the first part of the results would be useful to have, together with SD of the different subtests , also the range of values, moreover I think that raw scores are not informative about the level of performance because they are not scaled for age. For that reason, I suggest to the authors to show scaled score rather than raw scores.

# Figure 2: I suggest to the authors to make Axis in figure 2 squares to have a better visualization of the correlation. Additionally In the figure 2 R stands for r-pearson ? if it is the case I suggest to use lowercase letter.

# Looking in figure 2 is possible to identify subjects that showed a Verbal IQ below normality range (below scaled scores of 80), indeed this is reflected on the Total IQ scaled scores there are some subjects that showed an overall scores below 80. I therefore suggest that the authors consider eliminating these subjects because they have IQs at the lower end of the norm and could generate a bias in the results.

# Again for the analysis performed on the effect of degrees on IQ , the authors should use scaled scores rather than raw. Are there difference in visual orientation discrimination threshold across the different type of degrees?

# In order to give the reader a clearer and more comprehensive view of the results, I suggest the authors use a table containing the results of the multiple regression. Following this suggestion would be useful have an information of the amount of variance of Visual Orientation Threshold explained by the IQ subtests, so I would suggest to specify it in the presentation of the results.

Reviewer #2: The authors of this interesting and well written article investigate the relationship between a simple visual discrimination task, which should be indicative of the functioning of low levels of vision, and the performance obtained on an intelligence test, in its verbal and non-verbal components. The article reveals a certain correlation between the two indicators and provides a broad discussion on what may be the effects of other factors not necessarily linked to the variables taken into consideration, such as working memory.

I think the work can be accepted after major revision, which I indicate below.

1) Visual discrimination task. Authors should better specify why they chose this specific test. The presentation time of the two gabors is quite long, and may allow the subjects to make eye movements: what reason, procedural or theoretical, led them to choose this time? Also, were the location and interval of the standard stimulus and test stimulus randomized on a trial-trial basis? Probably the authors could write a more accurate caption of figure 1 where to insert the various information.

2) The authors, before proceeding to describe the results, should insert a brief description of the statistical analysis carried out, specifying for example that they used the Pearson test which does not assume the normality of the data.

3) In any case, I would also report whether or not the data obtained are normally distributed.

4) The correlations should be made, in my opinion, on the normalized data and not on the raw ones.

5) Figure 2 should show normalized data and the axes should be squared.

6. PLOS authors have the option to publish the peer review history of their article (what does this mean?). If published, this will include your full peer review and any attached files.

Reviewer #1: No

Reviewer #2: **Yes: **Nicola Megna

---

## [Author Response · Author response to Decision Letter 0]

30 Jun 2023

We would like to thank both reviewers for their positive attitude and insightful feedback regarding our manuscript. We have made amendments to take care of all the issues raised and due to this we believe that the manuscript is now at a better shape than it was at the initial submission.

Kind Regards,

Dr. Kyriaki Mikellidou

Reviewer #1: Evaluation

In this paper Mikellidou et al. investigate whether a basic visual function, such as visual orientation discrimination, can be correlated with high-order cognitive abilities measured through the Intelligent Quotient (IQ). The authors found a negative correlation between these measures, specifically lower precision in visual orientation discrimination was associated with lower level in particular subtests of Verbal and Non-Verbal IQ. Finally they conclude that these association are not causative but they demonstrate the relationship between basic sensory processing that could in some way influence the develop of individual cognitive factors .

The study appears to have been well executed and the writing is clear and communicates the main points well. I really appreciated the clarity of the introduction. I think that some points should be addressed or clarified before the study is suitable for publication.

Methods

# Visual Discrimination task : Is the position of the standard stimulus randomized across different trials, in order to avoid sequential order effects?

Yes, the standard stimulus was randomly presented either first or second to avoid sequential order effects and we now specify this in the text.

# why the authors choose to present the stimuli in periphery rather than in fovea? This for my opinion would be a way to avoid eye movements of the subjects.

This is exactly the reason; we wanted to avoid excessive eye movements and we know clarify this in the text (page 4)

Results

# in the first part of the results would be useful to have, together with SD of the different subtests , also the range of values, moreover I think that raw scores are not informative about the level of performance because they are not scaled for age. For that reason, I suggest to the authors to show scaled score rather than raw scores.

As suggested by the reviewer, we are now reporting scaled values and their range in the first part of the results.

# Figure 2: I suggest to the authors to make Axis in figure 2 squares to have a better visualization of the correlation. Additionally In the figure 2 R stands for r-pearson ? if it is the case I suggest to use lowercase letter.

We have made the axis square for better visualisation and have also changed R to r, in order to indicate Pearson’s r.

# Looking in figure 2 is possible to identify subjects that showed a Verbal IQ below normality range (below scaled scores of 80), indeed this is reflected on the Total IQ scaled scores there are some subjects that showed an overall scores below 80. I therefore suggest that the authors consider eliminating these subjects because they have IQs at the lower end of the norm and could generate a bias in the results.

We understand why the reviewer is raising this point, and while we did notice a few participants with scores below normality when we checked for outliers using a standardised method (values that are 3 standard deviations above or below our sample mean, so anything between 66-131) all of our datapoints fell between this range. In fact, our lowest value is 74.5 and our highest is 122. In order to avoid any perceived “cherry-picking” from our side, we opted to leave the dataset as is.

# Again for the analysis performed on the effect of degrees on IQ , the authors should use scaled scores rather than raw. Are there difference in visual orientation discrimination threshold across the different type of degrees?

We now use scaled scores to assess differences between degrees and also report that there are no differences in orientation discrimination threshold between the two types of degrees.

# In order to give the reader a clearer and more comprehensive view of the results, I suggest the authors use a table containing the results of the multiple regression. Following this suggestion would be useful have an information of the amount of variance of Visual Orientation Threshold explained by the IQ subtests, so I would suggest to specify it in the presentation of the results.

We now present the results of multiple regression in a table and clarify that the amount of variance explained by the two significant IQ tests is 15%.

Reviewer #2: The authors of this interesting and well written article investigate the relationship between a simple visual discrimination task, which should be indicative of the functioning of low levels of vision, and the performance obtained on an intelligence test, in its verbal and non-verbal components. The article reveals a certain correlation between the two indicators and provides a broad discussion on what may be the effects of other factors not necessarily linked to the variables taken into consideration, such as working memory.

I think the work can be accepted after major revision, which I indicate below.

1) Visual discrimination task. Authors should better specify why they chose this specific test. The presentation time of the two gabors is quite long, and may allow the subjects to make eye movements: what reason, procedural or theoretical, led them to choose this time? Also, were the location and interval of the standard stimulus and test stimulus randomized on a trial-trial basis? Probably the authors could write a more accurate caption of figure 1 where to insert the various information.

We now explain why we use a long(ish) presentation time; this had to do with the fact that our sample comprised inexperienced participants which found it challenging to carry out the task with shorter presentation times. We also expanded the figure legend to enhance clarity.

2) The authors, before proceeding to describe the results, should insert a brief description of the statistical analysis carried out, specifying for example that they used the Pearson test which does not assume the normality of the data.

We now have a small subsection “Data analysis” which provides this information.

3) In any case, I would also report whether or not the data obtained are normally distributed.

We thank the reviewer for this comment and we now report that some of the variables are not normally distributed and this is why we use statistical tests that do not require the assumption of normality. For the comparison of the degrees we now report Mann-Whitney U results because of this (please note that no changes have been observed from our initial submission).

4) The correlations should be made, in my opinion, on the normalized data and not on the raw ones.

The correlations are now performed on the normalised data.

5) Figure 2 should show normalized data and the axes should be squared.

Figure 2 does show normalised data and the axes are now squared for better visualisation.

---

## [Decision Letter · Decision Letter 1]

24 Jul 2023

Visual orientation discrimination skills are tightly linked with specific aspects of human intelligence

PONE-D-22-29542R1

Dear Dr. Mikellidou,

We’re pleased to inform you that your manuscript has been judged scientifically suitable for publication and will be formally accepted for publication once it meets all outstanding technical requirements.

Kind regards,

Nicola Megna, M.D.

Academic Editor

PLOS ONE

Additional Editor Comments (optional):

Reviewers' comments:

Reviewer's Responses to Questions

**Comments to the Author**

1. If the authors have adequately addressed your comments raised in a previous round of review and you feel that this manuscript is now acceptable for publication, you may indicate that here to bypass the “Comments to the Author” section, enter your conflict of interest statement in the “Confidential to Editor” section, and submit your "Accept" recommendation.

Reviewer #1: All comments have been addressed

Reviewer #2: All comments have been addressed

2. Is the manuscript technically sound, and do the data support the conclusions?

Reviewer #1: Yes

Reviewer #2: Yes

3. Has the statistical analysis been performed appropriately and rigorously? 

Reviewer #1: Yes

Reviewer #2: Yes

4. Have the authors made all data underlying the findings in their manuscript fully available?

Reviewer #1: Yes

Reviewer #2: Yes

5. Is the manuscript presented in an intelligible fashion and written in standard English?

Reviewer #1: Yes

Reviewer #2: Yes

6. Review Comments to the Author

Reviewer #1: the authors did a great job in responding to all the comments, now the paper is clear and presents its results concisely.

Reviewer #2: (No Response)

7. PLOS authors have the option to publish the peer review history of their article (what does this mean?). If published, this will include your full peer review and any attached files.

Reviewer #1: **Yes: **MARCO TURI

Reviewer #2: **Yes: **Nicola Megna

---

## [Editor Report · Acceptance letter]

31 Jul 2023

PONE-D-22-29542R1 

Visual orientation discrimination skills are tightly linked with specific aspects of human intelligence 

Dear Dr. Mikellidou:

I'm pleased to inform you that your manuscript has been deemed suitable for publication in PLOS ONE. Congratulations! Your manuscript is now with our production department. 

Kind regards, 

on behalf of

Dr. Nicola Megna 

Academic Editor

PLOS ONE